# Aphrophoridae as Potential Vectors of *Xylella fastidiosa* in Tunisia

**DOI:** 10.3390/insects14020119

**Published:** 2023-01-24

**Authors:** Sonia Boukhris-Bouhachem, Rebha Souissi, Raied Abou Kubaa, Maroun El Moujabber, Vladimir Gnezdilov

**Affiliations:** 1INRAT-National Agricultural Research Institute of Tunisia, Rue Hedi Karray, University of Carthage, Tunis 1004, Tunisia; 2CNR, Instituto per la Protezione Sostenibile delle Piante, Sede Secondaria di Bari, 70126 Bari, Italy; 3CIHEAM-Mediterranean Agronomic Institute, 70010 Bari, Italy; 4Zoological Institute, Russian Academy of Sciences, 1 Universitetskaya Emb., 199034 Saint Petersburg, Russia

**Keywords:** Auchenorrhyncha, morphology, *Xylella fastidiosa*, vectors, nymph, seasonal occurrence, forests, dry grassland, fruits-olive orchards, herbaceous plants

## Abstract

**Simple Summary:**

The bacterium *Xylella fastidiosa* induces many plant diseases causing yield losses and plant death. It is passively delivered into the xylem sap by spittlebugs vectors. These insects are small hemipterans of the Aphrophoridae family mostly ranging from 7 to 9 mm in length. They are quite polyphagous, sucking xylem sap from a multitude food plant species (spontaneous, ornamental and cultivated) present in forest, dry grassland and fruit orchards. Four spittlebug species naturally occur in Tunisia. Two species, *Philaenus tesselatus* and *Neophilaenus campestris,* seem to be potential vectors. Consequently, the risk of spreading the bacteria is important because of the exchanges between countries. Knowledge of the vector will enforce the available measures against plant pathogen invasion and help to control plant importations from infected countries.

**Abstract:**

The present study is an update on the situation of potential vectors of *Xylella fastidiosa* in Tunisia. Investigations in nine Tunisian regions (Nabeul, Bizerte, Béja, Jendouba, Zaghouan, Kairouan, Ben Arous, Tunis and Manouba) from 2018 to 2021 allowed for the observation of 3758 Aphrophoridae among a total of 9702 Auchenorrhyncha individuals collected by sweep net. Four Aphrophoridae species were identified with *Philaenus tesselatus* as most abundant (62%), followed by *Neophilaenus campestris* (28%), *Neophilaenus lineatus* (5%) and *Philaenus maghresignus* (5%). Aphrophoridae individuals were found to be particularly abundant in both forests of Nabeul and Jendouba, secondarily in olive groves and dry grassland. Furthermore, their distribution on weed hosts was followed in these two regions where nymphs and adults are widely distributed. *P. tesselatus* appears to be the most abundant species as determined either by conventional sweep netting for adults or by plant sampling on *Sonchus, Smyrnium, Cirsium, Rumex, Polygonum* and *Picris* for nymphs. Limited numbers of adults of *P. maghresignus* were detected by sweep netting, while nymphs of this species were found on *Asphodelus microcarpus* only. *N. campestris* was found in high numbers on plants belonging to the Poaceae family in forests, dry grassland and olive groves whereas *N. lineatus* occurred on herbs under or near olive trees and in dry grasslands.

## 1. Introduction

Sharpshooters (Cicadellinae) and a few other Auchenorrhyncha xylem feeders are global pests because they transmit the xylem-inhabiting bacterium *Xylella fastidiosa* (Xf) (Wells et al., 1987). This plant-pathogenic bacterium is the cause of yield losses and death of many economically important crops: grapevine, peach, plum, almond, *Citrus* spp., coffee, pecan and olive trees. It also infects natural and ornamental perennial plants, e.g., elm, sycamore, oak, maple, oleander, mulberry, ivy, platan, alfalfa and periwinkle. Their diseases take different names as Pierce’s disease [1], *Phony peach disease* (PPD) [2], *Citrus variegated chlorosis* (CVC) [3], Citrus X-disease, *Almond leaf scorch* (ALS) [4], *Plum leaf scald* (PLS) [5], *Blueberry leaf scorch* [6], *Oleander leaf scorch* (OLS) [7], *Coffee leaf scorch* (CLS) [8], *Mulberry leaf scorch* [9], *Alfalfa dwarf* [10], *Maple leaf scald* [11], *Elm leaf scorch* (BLS), *Periwinkle wilt* and *Pecan leaf scorch* [12] and *Olive quick decline syndrome* (OQDS) [13].

Furthermore, the bacterium has been found in many reservoirs of wild plants (often latently), such as grasses, sedges and trees [14,15]. This resulted in millions of dollars in damage to many fruits and horticultural crops [16,17].

*Xf* was introduced in Europe through the trade of asymptomatic coffee plants from Costa Rica (Central America) and has become an invasive pathogen in Europe. This bacterium may play a particular role in plant health, causing nonspecific water shortage symptoms or damages by plugging the xylem vessels with biofilm [18]. Vector–host–pathogen interactions determine whether an isolated pathogen outbreak will lead to settlement, persistence and a resulting epidemic development [19]. The bacterium uses part of the external cuticle of its vector as a nitrogen source employing enzymatic substances (chitinase) to dissolve exoskeleton [20]; as well, the vector ingests xylem sap for nitrogen and carbon sources [21]. The bacterium is efficiently acquired by insect vectors, with no latent period, and persists in infective adults indefinitely [22,23]. All xylem sap-feeding insects in Europe are considered to be potential vectors. The widespread polyphagous spittlebug *Philaenus spumarius* (Linnaeus, 1758) (Hemiptera: Aphrophoridae) has been identified as the main vector for *Xf* in the Apulian region of Italy [24]. In Italy *X. fastidiosa* subsp. pauca ST53 (Xfp53) has become a plague, causing extensive damages to production in olive oil orchards [25,26].

Few species of Aphrophoridae (Hemiptera) efficiently transmit the bacterium by feeding on xylem sap. *P. spumarius* and *Neophilaenus campestris* (Fall.) are confirmed vectors of *X. fastidiosa* ST53 in the Apulian region of Italy [27,28,29]. Nevertheless, the genus *Philaenus* Stål and *Neophilaenus* Haupt are widespread in the Palearctic and Nearctic regions [30].

*Philaenus* adults are hemipterans ranging from 5.3 to 6.9 mm in length and are reactive, suddenly jumping away if threatened. The adults are quite polyphagous, sucking xylem sap from a multitude of (exceeding 1000) food plant species [30,31,32,33]. The importance of *P. spumarius* was reported due to its role as a vector of *Peach yellow virus* [34] and Pierce’s disease “virus” in vines [35]. The second type of injury caused by the species is its directly harmful effect on plants by sucking. The nymphs of *P. spumarius* cause the main damage taking up to 280 times their own fresh weight of plant sap in 24 h [32].

From 1883 to 1884, Puton [36] collected *P. spumarius* and *N. campestris* in Tunisia near the Ain Draham site and Kessera site, respectively. Linnavuori [37,38] also collected *N. campestris* at Ain Draham, El Kef, Tunis, Bizerte, Raoued, Tabarka and Tebourba in 1962, and later in 1968, he reported *P. spumarius* at Bizerte. Melichar [39] described *P. tesselatus* based on Tunisian specimens. Beier and Wagner [40] questioned the status of *P. tesselatus,* recognizing it as a subspecies of *P. spumarius*. Halkka and Lallukka [41] supported Wagner’s opinion, and finally Nast [42] synonymized *P. tesselatus* with *P. spumarius*. Then, Drosopoulos and Remane [43] and Drosopoulos and Quartau [44] restored *P. tesselatus* as a valid species with the indication of differences in body size and male genitalia structure. However, recent genetic studies [33,45] showed no difference among these two species while currently their taxonomic positions remain based on genitalia morphology.

Five species of the genus *Neophilaenus* were reported from Tunisia [36,37,38,39,46,47]: *N. albipennis* (Fabricius, 1798), *N. campestris* (Fallén, 1805), *N. lineatus* (L., 1758), *N. longiceps* (Puton, 1895), *N. minor* (Kirschbaum, 1868).

Given the presence of several *Philaenus* and *Neophilaenus* species in Tunisia, it suggests the need for a better understanding of their assemblage as potential vectors of *Xf*. That knowledge will help in the tuning of the IPM strategies needed for the preventive protection [48,49] of the Tunisian economic orchards (olive, almonds, peach, grapevine, citrus, etc.) and other ornamental and forest trees in the case of unintentional introduction of *Xf*.

Here, we consider all the available data on the candidate vector species of spittlebugs found in the Tunisian context, based on data gathered in Aphrophoridae-promising areas from 2018 to 2021. Even if this study was partial in comparison to the whole territory, we aim to arrange and share the actual vector knowledge to help and strengthen the management strategy that prevent the pathogen introduction with imported infected plants.

## 2. Materials and Methods

### 2.1. Sampling Sites

Surveys for Aphrophoridae in Tunisia were carried out during the spring and summer seasons from 2018 to 2021. About 66 sites belonging to 9 Tunisian governorates (Figure 1) were surveyed (14 sites in Nabeul, 15 in Bizerte, 10 in Béja, 9 in Jendouba, 2 in Kairouan, 5 in Manouba, 2 in Tunis, 3 in Ben Arous and 6 sites in Zaghouan). Sampling was performed in different natural environments such as oak forests, herb layer understory and near/under cultivated olive and other fruit orchards (Table 1).

### 2.2. Sampling of Spittlebugs

The sampling of spittlebugs was carried out by sweep netting (conventional 38 cm diameter) for adults and by host plant sampling for nymphs. Sampling was performed in the spring and summer period between 9 h30 and 11 h30. Herbaceous vegetation (*Rosmarinus*, *Eryngium*, *Lavandula*, *Bellis*, *Dacus*, *Erica*, *Cistus*, *Rubia*, *Lathyrus*, *Rumex*, *Halimium,* etc.) in crop fields, shrubs and trees (*Pinus*, *Pistacia*, *Rubus*, *Myrtus*, *Quercus*, *Retama, Phillyrea*, *Juniperus*, *Nerium,* etc.) in forests were targeted for the collection of spittlebugs. Collection of adults was made taking care to sample exactly the same area on each occasion during two-hour catches consisting of 10 successive sweeps every 200 m. A total of 100 sweeps per sampling date and by site was performed. The same precaution was taken in other forest sites where several samples were obtained within a season. In fruit orchards and olive groves, only 30 sweeps were made (3 × 10 sweeps). Collected insects were conserved in falcon tubes and then transferred to the laboratory for incubation in the freezer through a 15 min treatment. Once killed, the different specimens were conserved in petri dishes until identification.

### 2.3. Morphological Identification

Individuals of Aphrophoridae were sorted out from other Auchenorrhyncha caught by sweeping and were counted. Identification to species level was based on male genitalia structure after performing the slide-mounting technique [50]. The terminalia were cut off and cleared for 20 min in warm 10% KOH water solution, bleaching all the tissues to leave the sole cuticle. Further dissection and cleaning were performed in distilled water before mounting in Essig’s Aphid Fluid [51]. Stereoscope scrutiny and identification were adopted following the specific keys [43,44,52,53]. A Leica MZ16 with IC3D camera and Leica DM5500B with Leica DFC490 camera were used to characterize adult specimens in dorsal, side and ventral view; the same cameras were used for imaging details of slide-mounted male genitalia. Dissected parts were dismantled from slides and preserved with glycerol in micro-vials with the corresponding adults in the collection of the Plant Protection Laboratory of Institut National de la Recherche Agronomique de Tunisie.

### 2.4. Distribution and Host Plant Colonization in 2021

This work also included some biological aspects based on natural plant cover in two regions, Nabeul (Dar Chichou) and Jendouba (Feija Forest and Ghardimaou) (Figure 1), where Aphrophoridae were collected in large numbers in the previous years (2018–2020). Twenty species of weeds with spittlebug nymphs were taken in March and April 2021 at Dar Chichou (17 March, 5 and 23 April) and on *Asphodelus microcarpus* at Feija (15 March, 8 and 30 April). Collected nymphs were maintained on their host plants and placed in water vials in individual plastic cages under controlled conditions of the growth chamber (25 ± 1 °C, 16:8 h). Host plants, localities and data on the specimen collections included in this study are shown in Figure 1. The daily observation of the nymphs allowed us to follow and count adult emergence for almost one month and one week.

## 3. Results

In the present work, a total of 9865 Auchenorrhyncha specimens were captured with 39.7% of them belonged to the Aphrophoridae family. In addition, nymphs were checked with plant sampling.

### 3.1. Spittlebug Identification and Description

Spittlebug species were collected from all prospected areas except for the Tunis regions and Soliman. A total of 3921 Aphrophoridae, 2082 females and 1839 males (47%), were captured. During the survey, four spittlebug species were reported from the nine visited regions of Tunisia: *Philaenus tesselatus* Melichar, 1899, *Philaenus maghresignus* Drosopoulos and Remane, 2000, *Neophilaenus campestris* and *Neophilaenus lineatus* (Figure 2).

#### 3.1.1. Philaenus Species

The two *Philaenus* species, *P. tesselatus* and *P. maghresignus*, were collected in several regions of Tunisia.

*Philaenus* adults from the dorsum view have a nearly oval body, an angularly convex anterior margin of vertex, front plate without median carina and hind tibiae bearing less than ten spurs apically (Figure 2). Observation of the aedeagus tip of *P. tesselatus* shows three pairs of aedeagal processes, one proximal, one medial and one distal, while the aedeagus tip of *P. maghresignus* is with only two pairs of processes, and the proximal processes are considerably longer than the distal ones.

Many publications support the regular presence of *P. spumarius* in the North Mediterranean countries [37,39,42,54,55]. *P. spumarius* appears to be absent from the Tunisian explored territories and was not collected in this four-year survey.

*P. tesselatus* was already reported in Tunisia since it is described by [39,40]. Later, this species was reported in two other Northern African countries, namely Morocco and Algeria [43,56] and in Southern European countries such as Spain [44] and Portugal [33,44,55]. This is the first report of *P. maghresignus* in Tunisia, which has a distribution apparently sympatric with *P. tesselatus*.

#### 3.1.2. Neophilaenus Species

The two *Neophilaenus* species, *N. campestris* and *N. lineatus*, were also identified. Both were already reported from Tunisia by [36].

*Neophilaenus* adults have a slender body in comparison to *Philaenus*, strongly convex anterior margin of vertex, front plate with a median carina, and hind tibiae with more than ten spurs. The *N. lineatus* head is acutely V-shaped. The forewings are with a whitish costal border and with a dark-brown longitudinal streak 2/3 of the wing length. The *N. campestris* head is less convex and with forewings often with two whitish spots at the border (Figure 2).

Other species of *Neophilaenus* were also previously reported from the country, but these were not captured in our survey.

### 3.2. Spittlebug Frequency According to Year and Region

Jendouba and Nabeul are the richest Aphrophoridae regions with a total of 770 and 766 individuals, respectively, followed by Bizerte and Béja. In Zaghouan, Kairouan and Manouba very few individuals were collected. Spittlebug populations in spring are much larger than in summer (Table 2). *P. tesselatus* is mostly present at Nabeul and Jendouba, *P. maghresignus* at Jendouba and Béja; *N. campestris* at Bizerte and Béja and *N. lineatus* at Bizerte and Nabeul.

### 3.3. Aphrophoridae Composition

*P. tesselatus* is the most abundant species, comprising about 62% of the total number of individuals, noting it as an endemic species. *N. campestris* (28%) follows; then *N. lineatus* and *P. maghresignus* show an equal rate of 5% (Figure 3).

### 3.4. Frequency Variations of Aphrophoridae Species According to Year

Globally, *P. tesselatus* was the most abundant species during all the prospected years with highest density in 2019 and 2021. *N. campestris* is the second most important species (Figure 4).

### 3.5. Distribution of Spittlebugs According to Land Type

Our results show that Aphrophoridae are the most abundant in forest with 74.1%, followed by fallow land and fruit orchards, particularly in olive groves (Table 3). *P. tesselatus* and *P. maghresignus* are almost exclusively present in forests. *N. campestris* occurs in forest, dry grassland and olive groves while *N. lineatus* occurs in fallow land and olive groves. Thus, *N. campestris* seems to be the potential vector in Tunisia.

### 3.6. Distribution and Breeding of Spittlebugs on Plants in Two Locations during 2021

Sampling was focused in two regions (Dar Chichou and Feija) for some biological aspects (host plants, nymph occurrence and adult emergence).

#### 3.6.1. Dar Chichou Forest

Nymphs were collected from ground vegetation in March and April; we did not observe them after that. *P. tesselatus* nymphs occurred on spontaneous plants such as *Sonchus oleraceus, Rumex* sp., *Cirsium arvense, Glebionis segetum* (syn. *Chrysanthemum segetum*)*, Smyrnium olusatrum*, *Rubia peregrina*, *Picris echioides*, *Scolymus grandiflorus,* etc., in this semi-natural forest (Figure 5). It is particularly common on *G. segetum* and *C. arvense.* The *P. tesselatus* nymph spends its life in a mass of froth on its food plants; after that, the herbaceous plants dried and we did not see foam anymore. The nymph samples checked in the laboratory took between 14 to 23 days to become adults. *G. segetum* was the most colonized host plant at Dar Chichou; 29% of *P. tesselatus* were counted there. This was followed by *Rumex* sp. (19%), *S. oleraceus* (18%), *C. arvense* (17%), *S. olusatrum* (7.5%), *P. echioides* (5%), *R. peregrina* (2.6%) and *S. grandiflorus* (1.3%).

Seven nymphs of *N. campestris* were collected only in a pool of twenty Poaceae plants composed of *Avena sativa*, *Bromus* sp., *Elymus* sp., *Holcus* sp., *Hordeum* sp., etc. Adults were obtained in the laboratory between 8 and 19 days.

Interestingly, *P. maghresignus* nymphs were not found on *Asphodelus* of the Dar Chichou site nor were *N. lineatus* between March and early May. Nevertheless, a few specimens of *P. maghresignus* were collected at the same site in July; they probably came from the neighboring *Asphodelus*.

In late spring and summer, the herbs dried, spittlebugs were captured by sweep netting with different frequencies on diverse trees and shrubs. Most of them were *P. tesselatus* on spontaneous plants, e.g., *Pinus*, *Pistacia*, *Rosmarinus*, *Rubus*, *Myrtus*, *Quercus*, *Eryngium*, *Retama*. *P. maghresignus* was rarely collected on shrubs and ground vegetation.

Furthermore, *N. campestris* was also collected in few numbers on grass vegetation near the forest. These numbers are limited comparatively to individuals captured under olive trees or dry grasslands in the Jendouba region or Bizerte.

#### 3.6.2. Feija Forest

*P. maghresignus* nymphs were collected in March and April. It colonizes *Asphodelus* plants from March to early May. Nymphs became adults in laboratory conditions in 16 to 22 days. In late April, we observed almost 40% of adults (Figure 6); afterwards the *Asphodelus microcarpus* dried and the adults went elsewhere. *P. maghresignus* adults were observed in high numbers on *Asphodelus* in late April and were rarely captured by sweeping in May and summer. Conversely, *P. tesselatus* nymphs were not found on ground vegetation in the forest. Curiously, many adults of this species were caught by sweep netting from May to August; they probably migrate from other sites.

In Tunisia, nymphs take about 3–4 weeks to become adult (from the first days of April until early May), and five nymph instars are observed in the laboratory developing within an enveloping foam. When several nymph individuals are present on the same host plant, they feed in shared bubble masses. Based on our rearing, the first newborn *P. tesselatus* and *P. maghresignus* appeared just when the nymph that molts to adult created the open window to exit in early May. Green individuals appeared from 5 May 2021 (Figure 7) and then individuals took different ornaments. A peak in hatching was noted on 23 and 26 April for *P. tesselatus* and *P. maghresignus*, respectively.

Both species *P. tesselatus* and *N. campestris* were widely present on herbaceous vegetation in both forests (Dar Chichou and Feija) and within the olive groves, respectively. The adults usually remained in the field while the food plants were available before the herbs declined. *N. campestris* can be commonly found in grasslands (Poaceae) and can also be found on trees, seeking shelter during hot days, for example on *Pinus, Pistacia, Rubus, Myrtus*, etc. *P. maghresignus* is associated with Liliaceae only and *N. lineatus* with Poaceae in dry grassland.

## 4. Discussion

-*P. tesselatus* occurs in six of the nine explored regions (Nabeul, Jendouba, Bizerte, Béja, Zaghouan, Kairouan) with the highest frequency in Jendouba and Nabeul. *P. tesselatus* is closely related to *P. spumarius*, differing by the structure of male genitalia [57]. This western Mediterranean species seems to be vicariant to *P. spumarius* in Tunisia and can be a candidate vector. Perhaps speciation might have started allopatrically in isolated populations of *P. spumarius* in northwest Africa where it is now absent or very limited and then spread north into the southern Iberian Peninsula in the same manner as *P. maghresignus* [43]. Such speciation needs a more favorable environment for *P. spumarius*, perhaps related to temperature within a climatic change context. It seems that the identity of *P. spumarius* and its presence in Tunisia is questionable, considering also the presence of *P. signatus* reported in the collection of Lindberg from Tunisia [54]. These two species were not found in the examined samples which remains unclear.-*P. maghresignus* is monophagous on *A. microcarpus* on which both nymphs and freshly emerged adults were present in high densities in spring. The catches of *P. maghresignus* were in low numbers in late spring and summer which does not reflect the field reality. This suggests the limit of the sweep net sampling for this species is in the underground vegetation. Accordingly, this method cannot give an absolute measure of the insect presence. Its status is not confirmed but *Philaenus italosignus* is very close to this species and it is a poor vector [29], hence it is not considered as high risk.-*N. campestris* occurred in seven regions of Tunisia and *N. lineatus* in six. *N. campestris* was frequent in our prospections. This species is able to transmit *X. fastidiosa* to olive trees in Italy [29]. In Spain it is considered as a serious threat to key crops that are vital for Spanish and French agriculture such as olive, almonds and grapevines [58,59,60]. In Portugal, both spittlebugs were the main species associated with olive groves [58]. In Turkey, *N. campestris* is one of the identified spittlebugs [61]. *N. campestris* and *P. spumarius* were recorded in very low numbers in Greece [62].-The status of *N. lineatus* as a vector is yet unknown.

During the spring, the situation in the Nabeul region showed that nymphs and adults of *P. tesselatus* were more abundant in forests (Dar Chichou) than olive groves and fruit orchards. *N. campestris* occurred in high numbers under olive groves and dry grassland while *N. lineatus* and *P. maghresignus* were rare. This study demonstrated that *P. tesselatus* could achieve their development on most of the Asteraceae, Polygonaceae and Apiaceae. Most of the foams encountered in early spring were almost entirely observed on species of these three families; other undetermined plants also showed foam, but in very low numbers that did not allow for sampling (less than 20). *P. tesselatus* has almost the same behavior as *P. spumarius,* described as highly polyphagous, with nymphs developing mainly on Asteraceae species, making a large distribution range possible.

Nymphs of *P. maghresignus* were abundant on *Asphodelus* in early spring at Jendouba (Feija Forest). However, summer collecting by sweep net showed that adults of *P. tesselatus* were the most abundant there while *P. maghresignus* was rare. Furthermore, *N. campestris* adults were numerous in spring on Poaceae under olive trees (Ghardimaou, Jendouba) and rare in summer. It was previously reported, when feeding opportunities decline, that the adults mass move to another available food source around, even to gymnosperms such as *Pinus* spp., *Cupressus* spp. or *Tuja* spp. [63]. Adults will re-enter the orchards in late August following spontaneous perennial herbs resprouting [32,64]. In late fall, adults return to the olive groves for oviposition. However, olive trees may act as transient hosts for spittlebugs and high population densities of these insect vectors should be avoided in areas where *X. fastidiosa* is present [58,65]. Interestingly, this situation is different in Tunisia where *P. tesselatus* is present in the forest whereas *N. campestris* seems to be circulating between forest, grassland and fruit orchards. *Philaenus* species were not abundant on olive canopies nor on herbaceous plants under olive trees but seems to occur on shrubs and weeds in forests. *Pinus* and *Pistacia* may host the spittlebugs in dry periods while Quercus species were found to host only Issidae during our prospections. This suggests migration out of the sampled zone over relatively longer distances or altitudes. It is possible that adults migrate to some humid zone in the neighboring valleys, as observed in Central Spain [58]. The movement of the spittlebugs between forest and crops is not very clear [65] and needs more observations, particularly for *N. campestris* which was encountered in many environments: forests, dry grassland and olive groves. This fact is different from Italy [66] and seems closer to the situation in Spain [58] and Morocco [67].

## 5. Conclusions

The survey activities of candidate vectors led to the identification of *P. tesselatus, P. maghresignus, N. campestris* and *N. lineatus*. A complex of candidate vectors was observed although *X. fastidiosa* is not currently present in Tunisia. *N. campestris* adults have a very low preference for deciduous trees and therefore the chances that they move to olive is rather low [68]. The high abundance observed (more than 200) of *N. campestris* in some prospected olive groves, suggests this spittlebug represents a threat if *X. fastidiosa* reaches this region. This would also threaten the surrounding cultures and forest trees. For *P. tesselatus*, no transmission test was conducted but this species is very close to *P. spumarius* and may be a good vector as well. Our findings show that *P. tesselatus* is comparable to *P. spumarius.* Both are polyphagous species that feed and reproduce on diverse plants in many habitats, including cultivated and non-cultivated hosts. *P. tesselatus* nymphs were detected on herbs in forests in April and May. They were present on most prospected weeds, which probably function as a reservoir for this species, namely Asteraceae plants. Nymphs from forests probably are not as harmful as adult vectors because they cannot move over distances [65]. Adults likely move from herbaceous plants to forest shrubs during August. Little is known about the mobility of *P. tesselatus* adults and their potential to colonize olive groves. *Neophilaenus* species were mostly collected on graminea with a variable frequency depending on the prospected area. *N. campestris* can be commonly found in grasslands but it can also be found on trees, seeking shelter during hot August days. In spite of the high abundance in dry grassland, this spittlebug was also encountered on weeds under olive trees as well as on forest shrubs. This observation is in accordance with a study in Portugal [69].

Additional surveys are needed to investigate other host plants of *P. tesselatus* and *N. campestris* to guide landscape management strategies targeting key reproductive and feeding hosts. These species have to be considered as a pest of main agricultural crops. Furthermore, it is important to know which environmental conditions influence the movement of vectors from other plants (in particular from herbaceous plants) to olive groves or fruit orchards. Considering the complex olive—*X. fastidiosa*—insect pathosystem, a different control approach is warranted, particularly including an examination of the role of natural predators in regulating vector population density and the use of essential oils to limit vector populations.

In the case of *X. fastidiosa* entrance in Tunisia, the presence of potential vectors will sustain the possible invasion by infecting and spreading the plant pathogen among susceptible plants, either cultivated or not. Knowing that transmission of *X. fastidiosa* is a very rapid process [70] and that spittlebugs are present, the risk is important to consider. It is, however, important to prevent the entry of infectious plants for planting from infected countries. The probability of entry of *Xf* to Tunisia through European exchanges where *Xf* is reported is very high with agricultural and ornamental plants.

## Figures and Tables

**Figure 1 insects-14-00119-f001:**
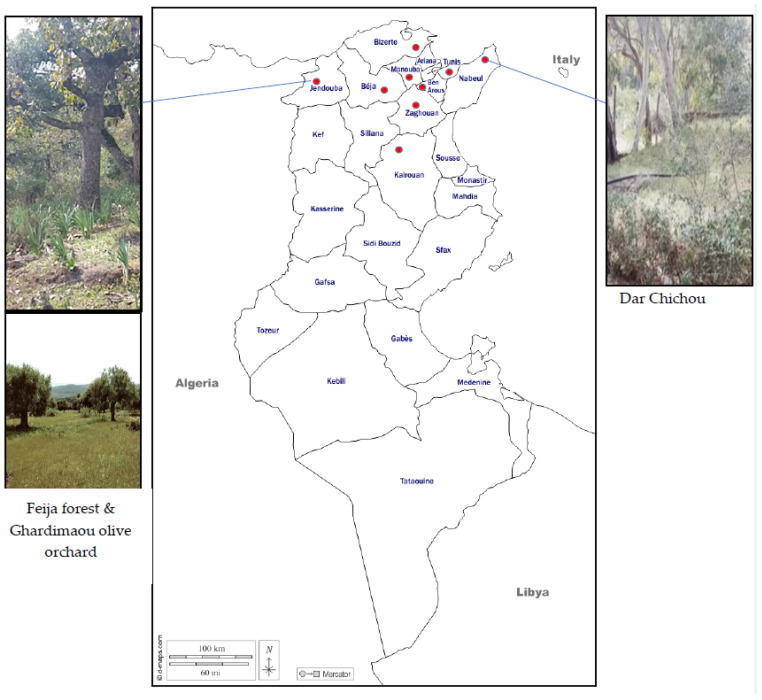
Spittlebug collecting regions (red circles) in Tunisia with focus on two reservoir forests: Dar Chichou and Feija.

**Figure 2 insects-14-00119-f002:**
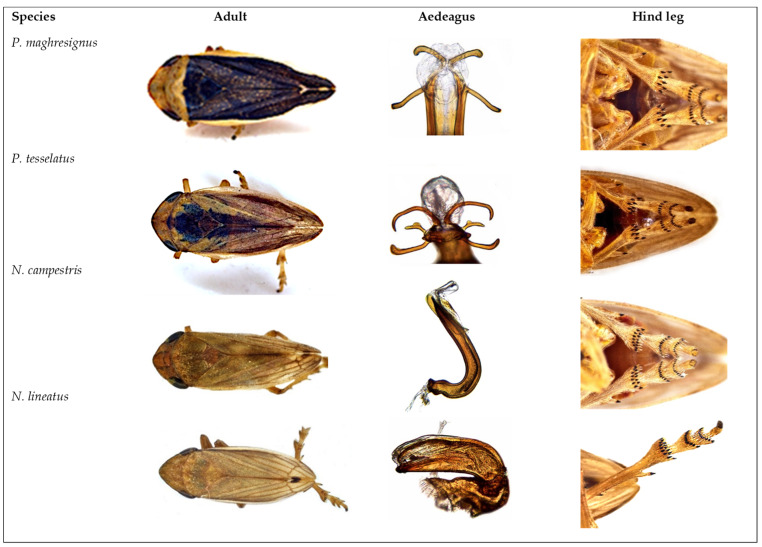
*Philaenus* and *Neophilaenus* species identification features (Adults, aedeagus and hind tibia).

**Figure 3 insects-14-00119-f003:**
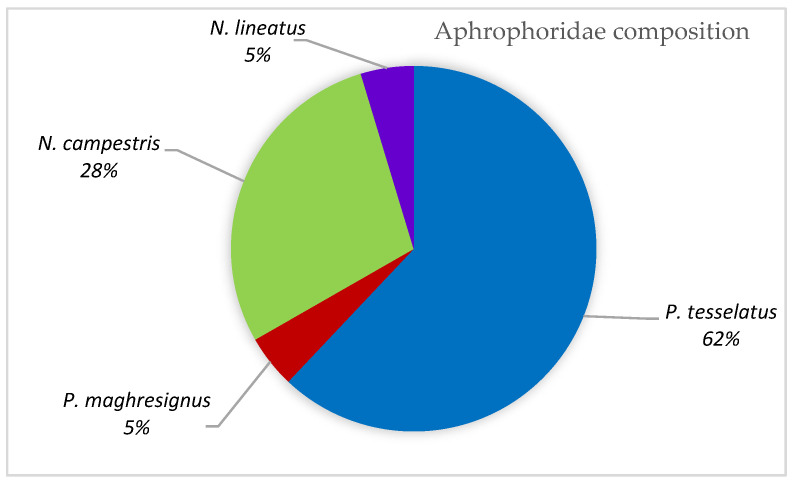
Composition of Aphrophoridae adults in Tunisian regions, 2018–2021.

**Figure 4 insects-14-00119-f004:**
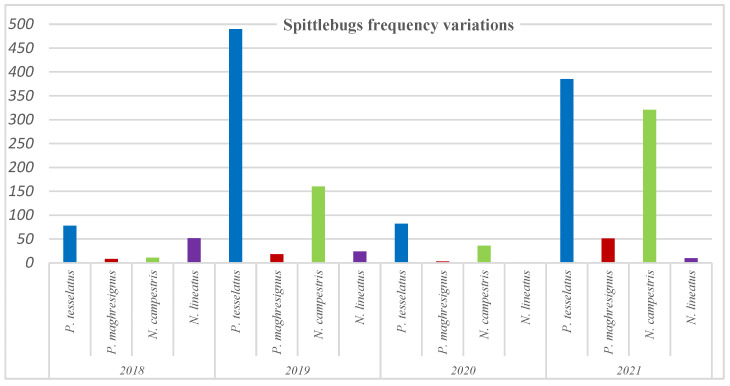
Frequency variations of the four spittlebugs according to year.

**Figure 5 insects-14-00119-f005:**
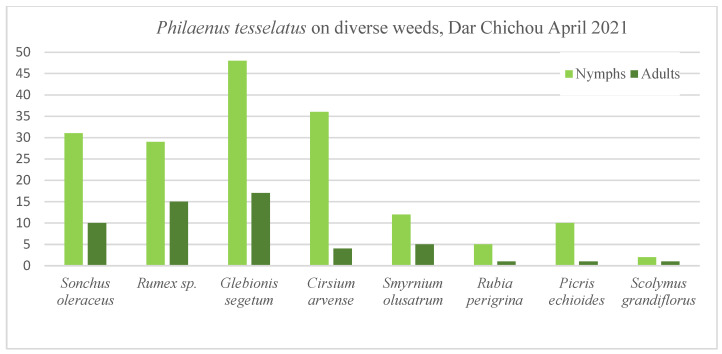
Spittlebug nymphs and adults counted on foliar sampling weeds at Dar Chichou, Tunisia.

**Figure 6 insects-14-00119-f006:**
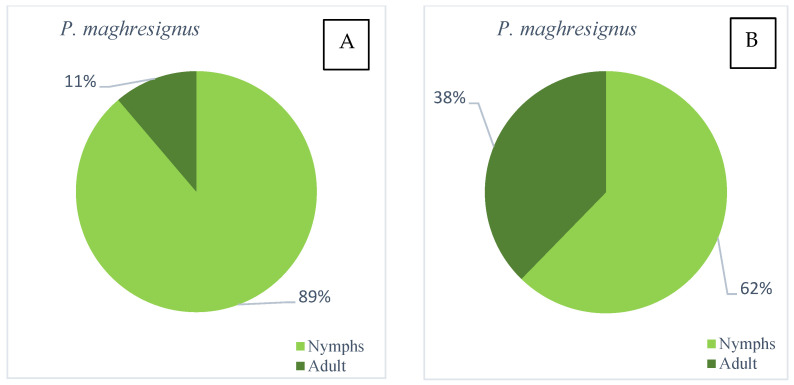
*P. maghresignus* nymphs and adults collected on *Asphodelus microcarpus*; (**A**), early April and (**B**), late April, Feija (Jendouba) 2021.

**Figure 7 insects-14-00119-f007:**
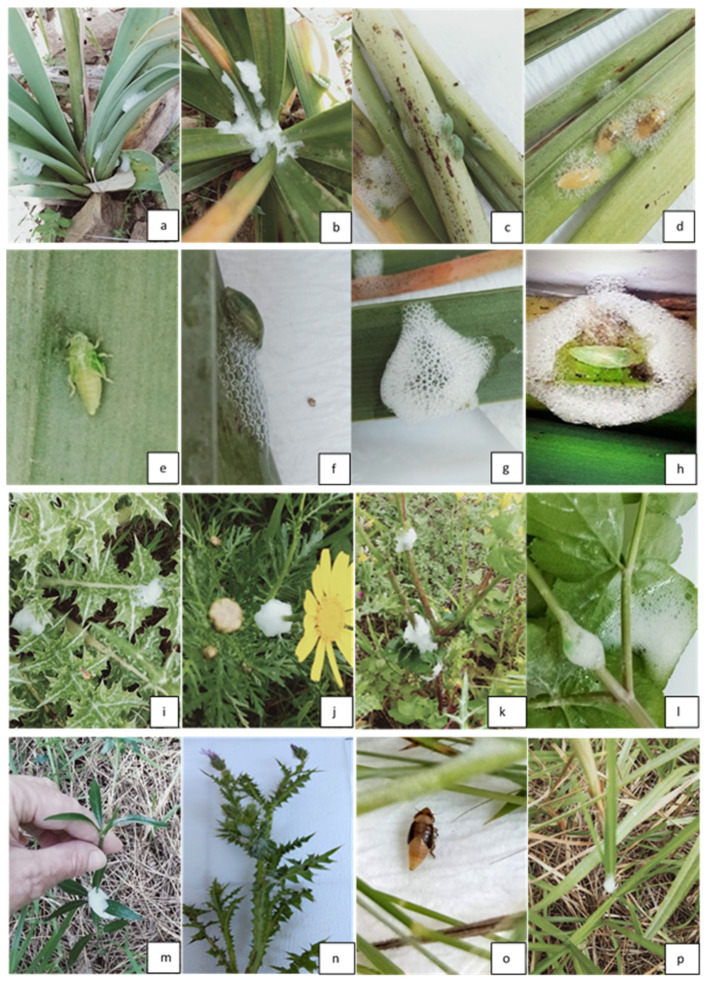
(**a**–**h**): Nymphs of *P. maghresignus* on Asphodelus plant collected from Feija; (**i**–**n**): host plants of *P. tesselatus* nymphs (Dar Chichou including Scolymus grandiflorus, Glebionis segetum, Sonchus oleraceus, Smyrnium olusatrum, Rubia perigrina, Cirsium arvense; (**o**,**p**): Nymph of *N. campestris* on Poaceae (Dar Chichou).

**Table 1 insects-14-00119-t001:** Prospected regions and collecting sites from 2018 to 2021 in Tunisia.

Regions	Forests/Natural Reserve	Dry Grassland	Fruit Orchards
Nabeul	Dar Chichou, Korbous, Kef Errand, Bou Argoub, Jebel Abderrahmen, Zougag	Douala, Brij, Haouaria, Tekelsa	Menzel Bouzelfa (olive), Douala (citrus and peach), Soliman (olive), Taffela (olive and grape vine)
Bizerte	Jebel Ichkeul and Lake, Rimel, Teskreya, Corniche	Khetmine, Pont de Bizerte, Utique, Alia, Sejnane	Mateur, Ain Ghlal (peach), Menzel Bourguiba (olive), Sidi Othman (olive), Azib (citrus)
Béja	Cap Negro, Chitana, Khroufa, Jebba	Nefza, Tebaba, Ain Essobh, Ouchtata	Jebba, Ouchtata (olive)
Jendouba	Feija, Ain Draham	Tabarka, Oued Snoussi, Balta, Fernana	Bousalem (peach, citrus, grape vine), Ghardimaou (olive), Ain Soltan (olive)
Zaghouan	Jebel Zaghouan, Oued el Galb	Fahs	Zriba, Fahs, Jebel el Ouest
Kairouan		Sidi Mahmoud-Oueslatia	Chebika (citrus, peach)
Ben Arous	Jebel Boukornine		Ben Arous (olive, grape vine), Mornag (peach)
Tunis	Belvédère park, Gammart		
Manouba		Batan, Saida, Jdaida	Mehrine (olive tree), Borj el Amri

**Table 2 insects-14-00119-t002:** Aphrophoridae species frequency (number of specimens) collected by sweep netting in Tunisia, 2018–2021.

Seasons	Males	Jendouba	Nabeul	Bizerte	Béja	Zaghouan	Kairouan	Manouba	Ben Arous	Tunis	T
Summer 2018	*P. tesselatus*	13	45	2	14	3	-	-	1	-	78
	*P. maghresignus*	-	7	-	1	-	-	-	-	-	8
	*N. campestris*	1	2	6	1	1	-	-	-	-	11
	*N. lineatus*	1	8	37	2	4	-	-	-	-	52
Spring 2019	*P. tesselatus*	70	326	6	4	1	1	-	-	-	408
	*P. maghresignus*	3	11	-	1	-	-	-	-	-	15
	*N. campestris*	-	7	112	2	1	2	-	-	-	124
	*N. lineatus*	-	-	18	-	-	-	-	-	-	18
Summer 2019	*P. tesselatus*	44	36	-	2	-	-	-	-	-	82
	*P. maghresignus*	4	-	-	-	-	-	-	-	-	4
	*N. campestris*	-	9	24	3	-	-	-	-	-	36
	*N. lineatus*	-	-	-	-	-	-	-	-	-	0
Summer 2020 *	*P. tesselatus*	137	40	2	9	-	-	-	-	-	188
	*P. maghresignus*	6	1	-	2	-	-	-	-	-	9
	*N. campestris*	28	2	1	3	-	-	-	-	-	34
	*N. lineatus*	-	-	-	-	-	-	-	-	-	0
Spring 2021	*P. tesselatus*	97	231	-	-	-	-	-	-	-	328
	*P. maghresignus*	18	5	19	-	-	-	-	-	-	42
	*N. campestris*	292	13	-	-	-	-	1	-	-	306
	*N. lineatus*	5	2	-	-	-	-	-	1	-	8
Summer 2021	*P. tesselatus*	38	14	-	5	-	-	-	-	-	57
	*P. maghresignus*	9	-	-	-	-	-	-	-	-	9
	*N. campestris*	5	8	1	1	-	-	-	-	-	15
	*N. lineatus*	-	-	7	-	-	-	-	-	-	7
Total		771	767	235	50	10	3	1	2	0	1839

***** No data for spring 2020 due to COVID-19 conditions.

**Table 3 insects-14-00119-t003:** Spittlebugs distribution.

Region	Forest	Dry Grassland	Fruit Orchards
*P. tesselatus*	99.8%	0	0.2% (2 grapevines)
*P. maghresignus*	100%	0	0
*N. campestris*	26.2%	22.7%	51.1% (olive)
*N. lineatus*	-	98.8%	1.2% (olive)
Total	74.1%	11.1%	14.8%

## Data Availability

The data presented in this study are contained within the article. All Aphrophoridae samples are conserved in the Plant Protection Laboratory of INRAT.

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
