# Peer review of "Aphrophoridae as Potential Vectors of Xylella fastidiosa in Tunisia"

_insects, 2023, doi:10.3390/insects14020119_

Round 1

Reviewer 1 Report

This article is about the distribution, habitats and host plants of four Aphrophoridae species in Tunisia, some of which are potential vectors of the phytopathogenic bacterium Xf, although this pathogen is not yet found in the country.

Although the research has a large sampling effort, the article has several writing and language style errors.

Page

Line

Comment

1

15

improve writing

20

delete

20

improve writing

33

Italics

49

Add References

2

50

improve writing

52

improve writing

53

delete

57

improve writing

58

delete

59

Delete

61

C

70

Add cite?

78

improve writing

81

Italics

81

Italics

83

Italics

83

Hemipterans (the term homoptera is not used anymore)

87

delete

87

improve writing

91-101

improve writing

103

Italics

105

improve writing

106

improve writing

3

110

improve writing

7

348

nine

349

Italics

351

Italics

10

365

improve writing

367

improve writing

368

improve writing

372

improve writing

373

improve writing

374

improve writing

375

improve writing

376

improve writing

377

improve writing

380

improve writing

12

398

improve writing

409

improve writing

410

Italics

417

improve writing

13

428

improve writing

438

improve writing

446

Italics

447

Italics

448

Italics

14

486

improve writing

493

it is colored

498

improve writing

499

improve writing

506

P.

507

Delete or improve writing

15

544

improve writing

16

546

improve writing

548

improve writing

551

improve writing

576

delete

589

improve writing

594

Italics

595

improve writing

Author Response

Dear Reviewer,

I improved the version according to your suggestions. 

Thank you for your help to get a better manuscript

Reviewer 2 Report

The manuscript describes the results from an inventory of four hemipteran species, which potentially may act as vectors for Xylella fastidiosa in Tunisia. If the disease will establish in Tunisia, it is valuable to know which vectors to look for in the different regions and habitats. Therefore, a revised version of the study will have both practical implications and be a contribution to the knowledge of species of this, sometimes neglected, group of insects. My main concern is how reliable the quantitative data and comparisons are. Based on the methods used, is it really possible to compare number between sites or years? To do that, the description of the methods has to be more detailed regarding the sweep netting. How many turns/strikes with the net per time unit or site? In order to be able to make the comparisons done in the ms and for possible future follow ups, the methods have to be described in such details and carefulness that someone else can repeat the procedure.  In addition to this, I have a number of issues to take care of:

Line 16: The regions are called habitats, which is wrong.

Many names of genera are not in italics as they should be.

The detailed description of the regions, lines 131-262, should go into an appendix or supplementary material. This is really not central for the study.

Lines 334-335 should go to section 2.2.

Table 3 should be reformatted into a figure instead and could be reduced to fit on one page.

I cannot find where figure 6 is referred to in the text.

The whole text needs a thorough revision of language.

Author Response

Dear Reviewer,

Thank you for the suggestion you send me. I do all the changes you request and I hope I succeed.

Sonia
